# Importance of Examining Incidentality in Vaccine Safety Assessment

**DOI:** 10.3390/vaccines12050555

**Published:** 2024-05-18

**Authors:** Yasusi Suzumura

**Affiliations:** YSP Medical Information Laboratory, Nagoya 468-0045, Japan; ysp@mbr.nifty.com

**Keywords:** incidentality, vaccine safety, adverse event, cohort study, self-controlled risk interval design, self-controlled case series method, COVID-19

## Abstract

The author believes that the principles of statistical methods for vaccine safety can be divided into three categories: comparison of adverse event incidence rates between vaccinated and unvaccinated groups, analysis of incidentality in the vaccinated group, and a combination of both. The first category includes the cohort study; the second, the self-controlled risk interval design (SCRI); and the third, the self-controlled case series method. A single *p*-value alone should not determine a scientific conclusion, and analysis should be performed using multiple statistical methods with different principles. The author believes that using both the cohort study and the SCRI for analysis is the best method to assess vaccine safety. When the cohort study may not detect a significant difference owing to a low incidence rate of an adverse event in the vaccinated group or a high one in the unvaccinated group, the SCRI may detect it. Because vaccines must have a higher level of safety than the pharmaceuticals used for treatment, vaccine safety is advisable to be assessed using methods that can detect a significant difference even for any value of the incidence rate of an adverse event. The author believes that the analyses of COVID-19 vaccine safety have areas for improvement because the proportion of papers that used the cohort study and the SCRI was negligible.

## 1. Introduction

The association between vaccines and adverse events should not be assessed arbitrarily; therefore, certain criteria are required. The World Health Organization (WHO) has published the assessment manual “Causality assessment of an adverse event following immunization (AEFI): user manual for the revised WHO classification (WHO manual) [1]”. This manual presents six criteria for assessment at the individual level: “temporal relationship, definitive proof that the vaccine caused the event, population-based evidence for causality, biological plausibility, consideration of alternative explanations, and prior evidence that the vaccine in question could cause a similar event in the vaccinee [1] (p. 10)”. Notably, population-based evidence, namely epidemiologic evidence, is needed even for assessment at the individual level.

Baker et al. described seven methods for examining population-based evidence [2]. The author believes that the principles of these statistical methods can be divided into three categories: comparison of adverse event incidence rates between the vaccinated and unvaccinated groups, analysis of incidentality in the vaccinated group, and a combination of both. The first category includes the cohort study, the case–control study, the case-centered analysis, the case-crossover analysis, and the risk interval design. The primary method for the first category is the cohort study, which compares the incidence rates of adverse events between the vaccinated and unvaccinated groups [2]. Some cohort studies have compared the incidence rates in the vaccinated group with estimated incidence rates based on historical cohorts using claim databases [3]. A significant difference in the incidence rates between the groups is detected if an association exists. The second category includes the self-controlled risk interval (SCRI) design. This design targets only the vaccinated group. The incidence rates of adverse events during the risk period (e.g., 1 to 21 days after vaccination) are compared with those during the control period (e.g., 22 to 42 days after vaccination) [2,4,5]. This design implicitly controls for time-invariant confounding factors such as age. If the occurrence of an adverse event is not incidental, a significant difference in the incidence rate by period is detected. The third category includes the self-controlled case series (SCCS) method. Traditionally, this method targets vaccinated and unvaccinated groups who have experienced a particular adverse event. The incidence rates of adverse events during the risk periods are compared with those during the control periods [2,6,7]. The control periods are set before and after vaccination in the vaccinated group. Furthermore, these are also set in the unvaccinated group. Thus, the SCCS has similar implications in that the incidence rates in the vaccinated group are compared with those in the unvaccinated group, as in the cohort study, and the incidence rates during the risk period are compared with those during the control period, as in the SCRI. This method implicitly controls for time-invariant confounding factors. A significant difference in the incidence rates by period is detected if an association exists.

This paper investigates how vaccine safety should be assessed and sheds light upon the importance and role of incidentality analysis.

## 2. Analysis Should Be Performed Using Multiple Statistical Methods

The American Statistical Association (ASA) statement asserted that a single *p*-value alone should not determine a scientific conclusion and that analysis should be performed using multiple statistical methods [8]. Further statistical analysis based on another principle is advisable, especially when no significant difference is found. This is because no significant difference cannot be interpreted as no association [9]; in other words, the association remains unclear. To elaborate further, no significant difference can be explained in two ways, as follows: there is no association, i.e., it is incidental; or there is an association but the incidence rate of an adverse event in the vaccinated group is very low, i.e., it is not incidental. A suitable explanation cannot be determined from a *p*-value based on the cohort study. Amrhein et al. recommend using confidence intervals instead of *p*-values for analysis [9]. However, even with this approach, determining whether an adverse event is incidental or not is difficult. This is because confidence intervals only provide a range of values, such as a risk ratio, without any evidence of whether it is incidental or not.

The author believes that when a significant difference is not found in the cohort study, it is advisable to conduct further analysis using the SCRI, which has a different principle. When the incidence rate of an adverse event in the vaccinated group is very low, no significant difference may be found in the cohort study, even if there is an association. No significant difference does not imply the absence of vaccine-related adverse events: it only indicates that their occurrence is not sufficiently large to yield statistical significance. On this occasion, a significant difference may be found in the SCCS or SCRI. This is because the SCCS may be advantageous when an adverse event is rare [2,10], and the SCRI may have the same advantage [11]. Because vaccines must have a higher level of safety than the pharmaceuticals used for treatment, vaccine safety is advisable to be assessed using methods that can detect a significant difference even if the incidence rate of an adverse event is low. If no significant difference is found in either the cohort study or the SCRI, the association remains unclear. However, conducting further analysis is practically difficult. The author believes that employing two methods with different principles is adequate for vaccine safety analysis. If a significant difference is found in the SCRI, it alone should not indicate vaccination discontinuation. The decision must be made comprehensively considering infection threat, vaccine efficacy, and safety.

In contrast, when the incidence rate of an adverse event in the unvaccinated group is very high (5 to 20%), no significant difference may be found in the cohort study, even if there is an association. This is because, assuming a 1% increase in the incidence rate due to a vaccine, the impact of a 1% increase in the unvaccinated group with an incidence rate of 15% is much smaller than that with an incidence rate of 0.1%. Thus, the cohort study may not detect a significant difference when the incidence rate of an adverse event in the vaccinated group is low or when high in the unvaccinated group. An example of a high incidence rate of an adverse event is spontaneous abortion [12,13]. Therefore, the author believes that assessing vaccine safety for this event is inappropriate based solely on the cohort study. When no significant difference is found on this occasion, it is advisable to assess vaccine safety using the SCRI in addition to the cohort study, and by considering the number of this event cases per million in the vaccinated group minus that in the unvaccinated group.

Whether a significant difference is detected depends on the incidence rate of an adverse event in the unvaccinated group. In other words, the higher the incidence rate, the less likely it is a significant difference would be detected. Each adverse event has a different incidence rate in the unvaccinated group. Therefore, a significant difference in the cohort study should be interpreted with an understanding of the above.

## 3. Comparing Sex Ratios by Period Is Another Incidentality Analysis

In addition to the SCRI, another method to examine incidentality is to compare sex ratios by period [11,14]. In this method, the sex ratios during the risk period are compared with those during the control period. Sex ratios are calculated by dividing the number of males by that of females and multiplying by 100. There is a high probability that the sex ratios during the risk period will be lower than those during the control period because of stronger immune responses to vaccines in women than in men [11,15] (Figure 1).

It is not appropriate to use the SCRI when reporting that bias exists. This bias occurs because the probability of reporting by physicians decreases with time after vaccination. In contrast, comparing sex ratios by period is less susceptible to reporting bias. This is because sex does not usually play a role in determining whether to report, regardless of the number of days after vaccination.

In a Japanese cohort study, no significant increase in the mortality rate due to coronavirus disease 2019 (COVID-19) vaccination was detected [16]. Even if the cohort study did not detect a significant difference, the SCRI or sex ratio comparisons by period may detect it. The author analyzed post-COVID-19 vaccination deaths by comparing the sex ratios by period in Japan [11]. A significant difference was detected in this study. This indicates that incidentality analysis may provide new evidence for an association between vaccination and deaths when the cohort study did not detect a significant difference.

## 4. The SCRI Should Not Be Viewed as a Subset of the SCCS

The SCRI and SCCS have self-controlled study designs. The SCRI has been described as a subset, a special case, or a variation of the SCCS [17,18,19]. However, the author believes that the SCRI should not be viewed as a subset of the SCCS but as an independent method for examining the incidentality of adverse events; conversely, the SCCS should be viewed as a statistical method that combines the cohort study and the SCRI.

The SCRI has some advantages over the SCCS. When unaware or difficult-to-correct biases exist between the vaccinated and unvaccinated groups in the cohort study, the SCCS is influenced by these biases, but the SCRI is not. This is because the SCCS targets the vaccinated and unvaccinated groups, whereas the SCRI targets only the vaccinated group. The limitation of the cohort study and the SCCS is that it is sometimes difficult to correct for some biases between the groups. From this perspective, when using the two methods for analysis, using the cohort study and the SCRI is recommended rather than using the cohort study and the SCCS. The SCRI also has the advantage of easier data collection than the SCCS because it targets only the vaccination group.

The author pointed to a disease severity bias as an example of bias that was difficult to correct in a cohort study in Japan [11,16]. Because COVID-19 vaccines are not recommended for patients in poor general condition [20], a disease severity bias exists between the vaccinated and unvaccinated groups. On this occasion, this bias may not have been properly corrected by adjusting it based on administrative claims data. This is because the claims data included only diagnoses and not disease severity in Japan.

## 5. Vaccine Safety Should Not Be Assessed Solely Based on Incidentality Analysis

The author believes that vaccine safety should not be assessed solely based on incidentality analysis of adverse events. This analysis may not provide conclusive evidence because it does not compare incidence rates with the unvaccinated group, and it is difficult to analyze delayed adverse events. Incidentality analysis is more valuable as a complementary method for the cohort study than as a standalone one. Furthermore, it is also valuable for signal detection. The WHO manual describes the signal as “The objective of signal evaluation is to draw conclusions on the presence or absence of a suspected causal association between an adverse event and a vaccine, and to identify the need for additional data collection or for risk minimization measures [1] (p. 12)”. The SCRI is used in the United States to detect signals in data analysis using the Vaccine Safety Datalink (VSD) [21]. The author believes that using both the cohort study and the SCRI for analysis is the best method to assess vaccine safety. When a significant difference is detected in the SCRI but not in the cohort study, it is advisable to assess vaccine safety by considering the number of adverse event cases per million in the vaccinated group minus that in the unvaccinated group.

## 6. Incidentality Analysis Requires a Graph Illustrating Days from Vaccination to a Particular Adverse Event and Case Count

The author noted the following: “To the author’s knowledge, the SCRI and the method of comparing sex ratios have not been discussed in terms of incidentality or even distribution” [11]. The author believes that the principle of the SCRI is to examine incidentality or even distribution. It is necessary to graph the number of days after vaccination to a particular adverse event and case count to examine this. If adverse events are not incidental (i.e., vaccine-related), the graph shows an uneven distribution; if they are incidental, the graph shows an even distribution (Figure 2).

The graph can be used to evaluate whether the risk period setting is appropriate. An inappropriate risk period setting leads to an incorrect result. For example, Klein et al. published a paper in which 23 serious adverse events were first analyzed using the SCRI with a risk period of 1 to 21 days after vaccination, and then only myocarditis was analyzed as a supplemental analysis with a risk period of 0 to 7 days [21] (Figure 3). A significant difference in myocarditis was detected only in the supplemental analysis. If this analysis was not performed, the risk of myocarditis could not be identified. The graph can prevent setting an inappropriate risk period. Furthermore, the absence of an explanation regarding the selection of the risk period for the initial analysis being 1 to 21 days instead of 0 to 20 days is problematic. The graph can provide clarification regarding which period is appropriate.

The WHO manual states, “At the individual level it is usually not possible to establish a definite causal relationship between a particular AEFI and a particular vaccine on the basis of a single AEFI case report [1] (p. 10)”. Therefore, assessing whether adverse events are incidental or vaccine-related at the individual level is usually impossible. This principle is important but has not always been followed in vaccine safety analysis. For example, Thomas et al. reported the results of a placebo-controlled pivotal efficacy trial evaluated through 6 months after BNT162b2 vaccination [22]. In Table S3 of the Supplementary Appendix, the numbers of adverse events were reported as follows: “Adverse Event (BNT162b2), Any event; 6617, Related; 5241”. “Related” is explained as “Assessed by the investigator as related to the investigational product”. According to the WHO manual, it is usually impossible and inappropriate for the investigator to determine these relationships. This paper did not describe the criteria used by the investigator to assess them. The details of the 1376 cases determined to be unrelated should be clarified. The only adverse events that can be determined to be unrelated are those for which other definite causes are found. Since mRNA vaccines are the first to be used in humans, it is unknown what adverse events may occur. The author believes that all adverse events other than those with other definite causes should have been analyzed in this paper, and the graphs should have been prepared to examine incidentality.

## 7. The Analyses of COVID-19 Vaccine Safety Have Areas for Improvement

The author believes that the analyses of COVID-19 vaccine safety have areas for improvement. These have been performed using a single statistical method in many papers. The PubMed search results were as follows: the cohort study, 276 [3,23,24,25,26,27,28]; the SCCS, 59 [29,30,31,32,33]; and the SCRI, 3 [21,34,35]. Four papers [16,36,37,38] using the cohort study and the SCCS, and three papers [39,40,41] using the cohort study and the SCRI have been published. As mentioned, the author believes that using the cohort study and the SCRI for analysis is the best method to assess vaccine safety. However, the proportion of papers that used these two methods was negligible (0.9%; 3/345). Thus, further studies should be conducted using these two methods. Papers with graphs illustrating the number of days from COVID-19 vaccination to a particular adverse event and case count were limited (11.6%; 8/69: the SCCS, 5 [29,30,31,32,33]; and the SCRI, 3 [21,35,41]). When graphs are not presented, evaluating whether the risk period setting is appropriate is impossible. The author believes that the presentation of graphs is essential in studies on the SCRI and SCCS analyses, particularly on the SCRI analysis.

All the results of the analyses have not always been published in the papers. The ASA statement recommended that all results be made public [8]. Klein et al. reported the results of supplementary analyses on myocarditis but not on the other 22 adverse events [21]. For a complete safety assessment, the author believes that it is advisable that the results of the supplemental analyses of all 23 adverse events are published in the Supplementary Appendix whenever possible. Failure to publish all the results analyzed to the possible extent may make it difficult to assess vaccine safety.

## 8. Comparison of Sex Ratios by Period Is Suitable for Vaccine Safety Assessment in Low- and Middle-Income Countries (LMICs)

Sisay et al. described a vaccine safety study in LMICs: “Major vaccination data sources were medical charts or self-reported cases based on clinical signs or symptoms [42]”. Correcting for reporting bias may be difficult in studies based on self-reported cases. They also described in the study, “Particularly, during the post-licensure phase and mass vaccination periods, simple and easily implementable study designs and methods may be preferred over more complex alternatives”. Analysis of sex ratio comparison by period can be performed using Fisher’s exact or chi-square tests [11]. As mentioned, this method is less susceptible to reporting bias. Therefore, the author believes that this method is suitable for vaccine safety assessment in LMICs. Additionally, this method can be used for signal detection in vaccine safety.

## 9. Evidence Is Necessary for Financial Support to Patients with Severe Adverse Events

In Japan, financial support has been provided to patients who experienced severe adverse events after COVID-19 vaccination [43]. As of 26 February 2024, JPY 46.7 million per person have been paid to the families of 459 dead patients after vaccination [44,45]. These included 118 patients with sudden deaths [45]. As sufficient medical findings were not reported in these sudden deaths and the causes of death were unclear, it is usually impossible to assess the association between death and vaccination. It is not scientific to provide financial support for deaths with unclear associations. This is because some deaths may have occurred incidentally. The author believes that support for sudden deaths is based on political considerations. However, financial support inherently requires evidence.

Japan lacks studies providing evidence of an association between adverse events and COVID-19 vaccination. Only one cohort study on COVID-19 vaccine safety in Japan has been reported [16]. If Japan is to conduct more studies in the future, the author believes that it is necessary to conduct not only a cohort study but also SCRI or sex ratio comparisons by period. Nevertheless, developing a system in Japan, such as the VSD in the United States, is necessary to minimize reporting bias when using the SCRI.

## 10. Conclusions and Future Directions

Analysis should be performed using multiple statistical methods based on different principles. Further statistical analysis based on another principle is advisable, especially when no significant difference is found. The author believes that using both the cohort study, which compares incidence rates, and the SCRI, which examines incidentality, is the best method to assess vaccine safety. When unaware or difficult-to-correct biases exist between the vaccinated and unvaccinated groups in the cohort study, the SCCS is influenced by these biases, but the SCRI is not. When the cohort study may not detect a significant difference owing to a low incidence rate of an adverse event in the vaccinated group or a high one in the unvaccinated group, the SCRI may detect it. Because vaccines must have a higher level of safety than the pharmaceuticals used for treatment, vaccine safety is advisable to be assessed using methods that can detect a significant difference even if the incidence rate of an adverse event is low in the vaccinated group or high in the unvaccinated group. When a significant difference is detected in the SCRI but not in the cohort study, vaccine safety should be assessed by considering the number of adverse event cases per million in the vaccinated group minus that in the unvaccinated group. The SCRI is more valuable as a complementary method for the cohort study than as a standalone one. Incidentality analysis, such as the SCRI, requires a graph illustrating the number of days from vaccination to a particular adverse event and case count. The graph can be used to evaluate whether the risk period setting is appropriate. An inappropriate risk period setting leads to an incorrect result.

The author believes that the analyses of COVID-19 vaccine safety have areas for improvement. These have been performed using a single statistical method in many papers. In contrast, the proportion of papers that used the cohort study and the SCRI was negligible. Furthermore, papers with graphs illustrating the number of days from COVID-19 vaccination to a particular adverse event and case count were limited. When graphs are not presented, evaluating whether the risk period setting is appropriate is impossible. All of the results of these studies have not always been published. Failure to publish all the results analyzed to the possible extent may make it difficult to assess vaccine safety. For further studies, the author believes that those using both the cohort study and the SCRI are advisable.

In addition to the SCRI, another method to examine incidentality is to compare sex ratios by period. This method is less susceptible to reporting bias. Therefore, this method is suitable for vaccine safety assessment in LMICs, where vaccination data sources may have reporting bias. Additionally, this method can be used for signal detection in vaccine safety. The SCRI and sex ratio comparisons by period may provide evidence for financial support when the cohort study fails to detect a significant difference.

Finally, the author hopes that this paper will lead to an increase in future studies using the cohort study and the SCRI.

## Figures and Tables

**Figure 1 vaccines-12-00555-f001:**
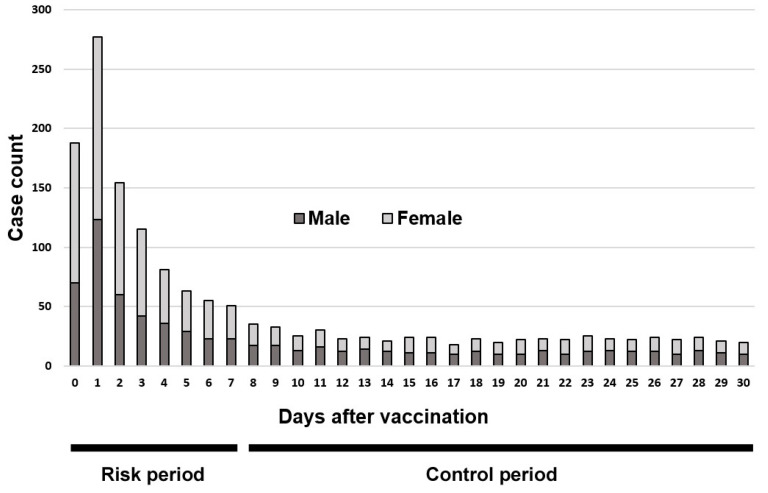
Artificial graph illustrating the number of days after vaccination to a particular adverse event and case count. The case count is sorted by sex: male (dark gray) and female (light gray). This graph shows that the sex ratio during the risk period was lower than that during the control period.

**Figure 2 vaccines-12-00555-f002:**
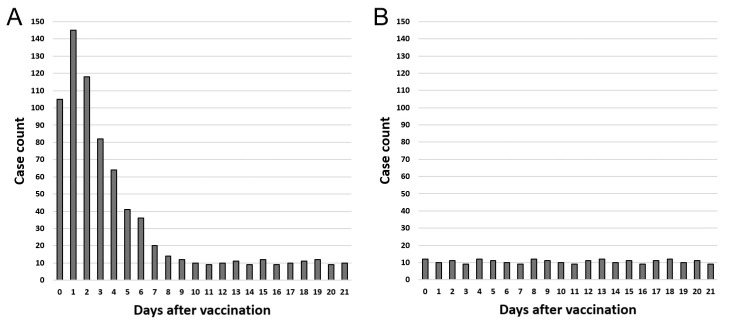
Artificial graph illustrating the number of days after vaccination to a particular adverse event and case count. Graph (**A**) shows an uneven distribution. Graph (**B**) shows an even distribution.

**Figure 3 vaccines-12-00555-f003:**
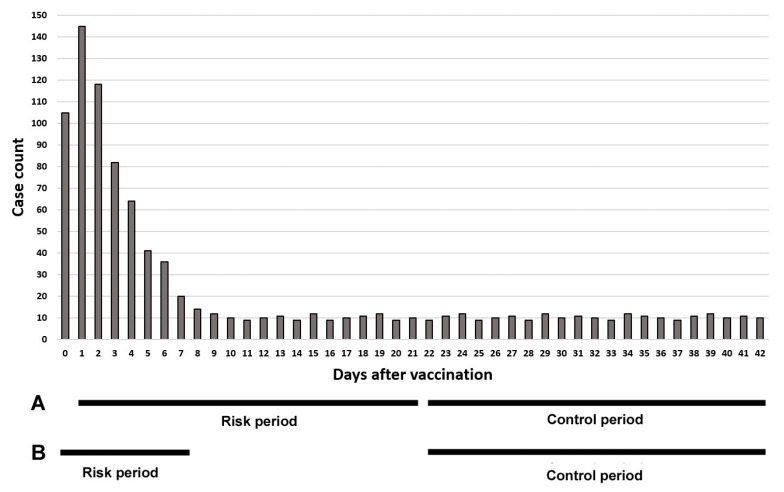
Artificial graph illustrating the number of days after vaccination to a particular adverse event and case count. Bar A shows a risk period of 1 to 21 days and a control period of 22 to 42 days. Bar B shows a risk period of 0 to 7 days and a control period of 22 to 42 days. In this graph, a risk period of 0 to 7 days is appropriate.

## Data Availability

Not applicable.

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
