# Peer review of "Importance of Examining Incidentality in Vaccine Safety Assessment"

_vaccines, 2024, doi:10.3390/vaccines12050555_

Round 1

Reviewer 1 Report

Comments and Suggestions for Authors

The manuscript is well-written and interesting. I recommend the publication.

The author addresses the examination of safety of vaccines, especially in the cases where either the background incidence of serious adverse events is high or the occurence of such events for the vaccinees is seldom, so the adverse events are often assessed as not significant. He emphasizes to additionally use SCRI or SCCS, which is currently done quite seldom (1.7% of published papers).

The author does not examine real data of adverse events of vaccinations so the only figure shows a diagram with artificially created random data to show the difference between the signal of adverse events caused by vaccination compared to adverse events caused mainly by background incidence. This is principially good to show the difference in signals - in reality you will not have such a unique signal within your data: You will have a combination of the (disturbed) signal of the AEs of the vaccination, and a decreasing number of reports of cases caused by the background incidence so often it will not be so easy to distinct between both cases.
So the manuscript emphasizes to improve the methods to examine AEs of vaccines, and shows some methods to do so, but it doesn't show the improvements with real data.

Author Response

I thank you for your valuable feedback.

Comment: The author does not examine real data of adverse events of vaccinations so the only figure shows a diagram with artificially created random data to show the difference between the signal of adverse events caused by vaccination compared to adverse events caused mainly by background incidence. This is principially good to show the difference in signals - in reality you will not have such a unique signal within your data: You will have a combination of the (disturbed) signal of the AEs of the vaccination, and a decreasing number of reports of cases caused by the background incidence so often it will not be so easy to distinct between both cases.
So the manuscript emphasizes to improve the methods to examine AEs of vaccines, and shows some methods to do so, but it doesn't show the improvements with real data.

Response : In a Japanese cohort study, no significant increase in the mortality rate due to coronavirus disease 2019 (COVID-19) vaccination was detected [16]. Even if the cohort study did not detect a significant difference, the SCRI or sex ratio comparisons by period may detect it. The author analyzed post-COVID-19 vaccination deaths by comparing the sex ratios by period in Japan [11]. A significant difference was detected in this study. This indicates that incidentality analysis may provide new evidence for an association between vaccination and deaths when the cohort study did not detect a significant difference.

The above was described in sections 8 and 9. I created a new section 3, "Comparing sex ratios by period is another incidentality analysis," and moved some sentences from sections 8 and 9 to section 3.

Reviewer 2 Report

Comments and Suggestions for Authors

This an interesting and important topic.

1. I have noted below paragraphs in which you provided details from studies to illustrate your arguments. Could you please increase the number of such illustrations and with more details and graphic presentations?

Lines 152-161: The graph can be used to evaluate whether the risk period setting is appropriate. An inappropriate risk period setting leads to an incorrect result. For example, Klein et al. published a paper in which 23 serious adverse events were first analyzed using the SCRI with a risk period of 1 to 21 days after vaccination, and then only myocarditis was analyzed as a supplemental analysis with a risk period of 0 to 7 days [19]. A significant difference in myocarditis was detected only in the supplemental analysis. If this analysis was not performed, the risk of myocarditis could not be identified. The graph can prevent setting an inappropriate risk period. Furthermore, the absence of an explanation regarding the selection of the risk period for the initial analysis being 1 to 21 days instead of 0 to 20 days problematic. The graph can provide clarification regarding which period is appropriate.

Lines 166-179: For example, Thomas et al. reported the results of a placebo-controlled pivotal efficacy trial evaluated through six months after BNT162b2 vaccination [20]. In Table S3 of the Supplementary Appendix, the numbers of adverse events were reported as follows: "Adverse Event (BNT162b2), Any event; 6617, Related; 5241." "Related" is explained as "Assessed by the investigator as related to the investigational product." According to the WHO manual, it is usually impossible and inappropriate for the investigator to determine these relationships. This paper did not describe the criteria used by the investigator to assess them. The details of the 1,376 cases determined to be unrelated should be clarified. The only adverse events that can be determined to be unrelated are those for which other definite causes are found. Since mRNA vaccines are the first to be used in humans, it is unknown what adverse events may occur. The author believes that all adverse events other than those with other definite causes should have been analyzed in this paper, and the graphs should have been prepared to examine incidentality.

Lines 207-213: In addition to the SCRI, another method to examine incidentality is to compare sex ratios by period [11,39]. In this method, the sex ratios during the risk period are compared with those during the control period. Sex ratios are calculated by dividing the number of males by that of females and multiplying by 100. There is a high probability that the sex ratios during the risk period will be lower than those during the control period because of stronger immune responses to vaccines in women than in men [11,40]. The author used this method to analyze post-COVID-19 vaccination deaths in Japan [11].

180

Lines 214-226: It is not appropriate to use the SCRI when reporting bias exists. This bias occurs because the probability of reporting by physicians decreases with time after vaccination. In contrast, comparing sex ratios by period is less susceptible to reporting bias. This is because sex does not usually play a role in determining whether to report, regardless of the number of days after vaccination. Sisay et al. described a vaccine safety study in LMICs: "Major vaccination data sources were medical charts or self-reported cases based on clinical signs or symptoms [41]." Correcting for reporting bias may be difficult in studies based on self-reported cases. They also described in the study, "Particularly, during the post-licensure phase and mass vaccination periods, simple and easily implementable study designs and methods may be preferred over more complex alternatives." Analysis of sex ratio comparison by period can be performed using Fisher's exact or chi-square tests [11]. Therefore, the author believes that this method is suitable for vaccine safety assessment in LMICs.

2. The RCT is the preferred study design for comparing outcomes. Could you please check if there are RCTs with enough participants to examine adverse vaccine events? I did a search in Medline (which I will send to the editor for transmission to you) using these search terms:

1     exp Vaccination/ (114078)

2     adverse events.mp. (218816)

3     randomised controlled trials.mp. (29614)

4     exp Randomized Controlled Trial/ (612834)

5     3 or 4 (641511)

6     1 and 2 and 5 (559) but the search did not download 599 but only downloaded 50 RCTs (see attached for literature search performed). Perhaps you might like to repeat or improve the search.

Please also note that this important RCT has just been published in Lancet Infectious Diseases 2024: 

Immunogenicity and safety of an ORF7-deficient skin[1]attenuated and neuro-attenuated live vaccine for varicella: a randomised, double-blind, controlled, phase 2a trial. Hong-Xing Pan*, Ling-Xian Qiu*, Qi Liang*, Zhen Chen*, Ming-Lei Zhang, Sheng Liu, Guo-Hua Zhong, Kong-Xin Zhu, Meng-Jun Liao, Jia-Lei Hu, Jia-Xue Li, Jin-Bo Xu, Yong Fan, Yue Huang, Ying-Ying Su, Shou-Jie Huang, Wei Wang, Jin-Le Han, Ji-Zong Jia, Hua Zhu, Tong Cheng, Xiang-Zhong Ye†, Chang-Gui Li†, Ting Wu†, Feng-Cai Zhu†, Jun Zhang†, Ning-Shao Xia

Author Response

I thank you for your valuable feedback. I have revised the manuscript accordingly.

Comment 1: I have noted below paragraphs in which you provided details from studies to illustrate your arguments. Could you please increase the number of such illustrations and with more details and graphic presentations?

Response 1: I have added two Figures.

Figure 1. Artificial graph illustrating the number of days after vaccination to a particular adverse event and case count. The case count is sorted by sex: male (dark gray) and female (light gray). This graph shows that the sex ratio during the risk period was lower than that during the control period.

Figure 3. Artificial graph illustrating the number of days after vaccination to a particular adverse event and case count. Bar A shows a risk period of 1 to 21 days and a control period of 22 to 42 days. Bar B shows a risk period of 0 to 7 days and a control period of 22 to 42 days. In this graph, a risk period of 0 to 7 days is appropriate.

Comment 2: The RCT is the preferred study design for comparing outcomes. Could you please check if there are RCTs with enough participants to examine adverse vaccine events?

Response 2: The RCT is the preferred study design for comparing outcomes. However, it is not suitable for analyzing adverse events with very low incidence rates. Rather, observational studies, such as the cohort study, are better suited to analyze them. This is because the sample size of observational studies is usually much larger than that of RCTs. I sincerely apologize; however, I believe it is unnecessary to check RCTs with enough participants in this paper.

Reviewer 3 Report

Comments and Suggestions for Authors

This paper reports the author's beliefs regarding principles of statistical methods for assessing vaccine safety. As such, it is well written, and I have no recommendations for revision. 

Author Response

Comment: This paper reports the author's beliefs regarding principles of statistical methods for assessing vaccine safety. As such, it is well written, and I have no recommendations for revision. 

Response: I thank you for your positive feedback.

Round 2

Reviewer 2 Report

Comments and Suggestions for Authors

Appropriate changes in the paper.